# Effects of Insular Cortex on Post-Stroke Dysphagia: A Systematic Review and Meta Analysis

**DOI:** 10.3390/brainsci12101334

**Published:** 2022-10-02

**Authors:** Jia Qiao, Zhimin Wu, Xue Cheng, Qiuping Ye, Meng Dai, Yong Dai, Zulin Dou

**Affiliations:** 1Department of Rehabilitation Medicine, The Third Affiliated Hospital of Sun Yat-sen University, Guangzhou 510630, China; 2Department of Neurosurgery, The Third Affiliated Hospital of Sun Yat-sen University, Guangzhou 510630, China; 3Department of Rehabilitation Medicine, The First Affiliated Hospital of Sun Yat-sen University, Guangzhou 510080, China; 4Clinical Medical College of Acupuncture, Guangzhou University of Chinese Medicine, Guangzhou 510006, China

**Keywords:** post-stroke dysphagia (PSD), aspiration, swallow–breathing coordination, insular cortex, systematic review

## Abstract

Objective: To investigate the relationship of lobar and deep brain regions with post-stroke dysphagia (PSD). Method: The databases of Medline, Embase, Web of Science, and Cochrane Library were searched from the establishment to May 2022. Studies that investigated the effects of lesions in lobar and deep brain regions on swallowing function after stroke were screened. The primary outcomes were PSD-related brain regions (including aspiration-related and oral transit time-related brain regions). The secondary outcomes were the incidence rate of PSD. The brain regions with the most overlap in the included studies were considered to be most relevant to PSD, and were presented as percentages. Data were compared utilizing the *t*-tests for continuous variables and χ^2^ for frequency-based variables. Result: A total of 24 studies and 2306 patients were included. The PSD-related lobar and deep brain regions included the insular cortex, frontal lobe, temporal gyrus, basal ganglia, postcentral, precentral, precuneus, corona radiate, etc. Among these brain regions, the insular cortex was most frequently reported (taking up 54.2%) in the included studies. Furthermore, the total incidence rate of PSD was around 40.4%, and the incidence of male was nearly 2.57 times as much as that of female (χ^2^ = 196.17, *p* < 0.001). Conclusions: In lobar and deep brain regions, the insular cortex may be most relevant to PSD and aspiration, which may be a potentially promising target in the treatment of PSD.

## 1. Introduction

Strokes are common diseases; in the aggregate, they are among the leading causes of mortality and long-term disability in developed countries, and their incidence is increasing as the population ages. There are an estimated 100 million individuals living with stroke sequelae worldwide, which has an enormous impact on patients’ quality of life and raises the financial burden of medical treatment [1]. Post-stroke dysphagia (PSD) is a major complication or significant sequelae after stroke, which may manifest as aspiration, pharyngeal residual, delayed swallowing initiation, etc., and be closely related to aspiration pneumonia, starvation, and dehydration [2,3,4]. Early data suggest that it affects 29–78% of patients with stroke and is linked to a higher risk of hospital readmissions and death [5,6]. Besides, other PSD-related factors include advanced age, unilateral spatial neglect (USN), etc., because of the decline of physiological functions [7,8].

The medulla oblongata is usually considered to be the swallowing center, where the swallowing central pattern generator (swCPG) participates in the coordinated contraction of lingual, pharyngeal, and laryngeal muscles [9,10]. However, the lobar and deep brain regions also play an important role in the neural regulation of swallowing (including prefrontal lobe, temporal lobe, insular lobe, parietal lobe, thalamus, supplementary motor area, corona radiata, internal capsule, periventricular white matter, etc.). Any lesions of these brain regions are thought to be related to PSD [11,12,13,14]. For example, the activation of the primary sensory/motor cortex is believed to predominate in reflex swallowing [15,16,17], while the activation of the primary sensory/motor cortex, insular, prefrontal lobe, subgenual cingulate gyrus, cuneate, and precuneus is associated with spontaneous swallowing [11,13]. The aspiration, a major complication of PSD, has been reported to be related to lesions of brainstem [14]. However, whether there is an association between aspiration and lesions of lobar and deep brain regions has not been well investigated.

Neuroimaging technologies, including functional magnetic resonance imaging (fMRI), make it possible to identify the relationship between brain regions and PSD [18,19,20]. MRI provides high sensitivity and specificity for ischemic stroke, in which the diffusion-weighted imaging (DWI) is most sensitive to hyperacute cerebral infarction [21,22]. Besides, voxel-based lesion-symptom mapping (VLSM) is a technique used to make voxel-wise statistical comparisons between lesion sites and neuropsychological test performance [11]. Furthermore, videofluoroscopic swallowing study (VFSS) and fiberoptic endoscopic evaluation of swallowing (FEES) are widely used in evaluation of PSD, and both VFSS and FEES are characterized as the gold standards for assessing PSD with different advantages and indications [23].

Several studies have reported that neuroanatomical location is linked to the incidence, severity, and characteristics of PSD [13,17,24,25]. Given that the lobar and deep brain regions related to PSD are reportedly different, we conducted this systematic review and compared existing studies in the literature. The aim of this study was to investigate the relationship of lobar and deep brain regions with PSD. We hope to provide evidence that can guide treatment for PSD in clinical practice.

## 2. Method

### 2.1. Data Sources and Searches

Medline, Embase, Web of Science, and the Cochrane Library were systematically searched from the establishment to May 2022. We manually searched the related studies with the MESH term “stroke” or “post-stroke” or “poststroke” or “hemiplegia” or “hemiparesis” or “paresis” or “paretic” or “hemipareticand” and “lesion” or “site” or “region” and “dysphagia” or “swallowing disorder” or “deglutition”. A hand search was additionally performed to screen the articles to further clarify potentially eligible studies in the pre-selected articles. The full search strategy and Mesh terms were detailed in the Appendix A. The retrieval was based on the subject terms, keywords, or titles. This study was registered with PROSPERO (CRD42022339058).

### 2.2. Inclusion and Excluded Criteria

Inclusion criteria were: (1) detailed brain regions related to stroke were confirmed by MRI or CT; (2) age >18 years old; (3) swallowing function was evaluated by VFSS, FEES, or clinical evaluations; (4) studies published in English; (5) case-control studies; (6) prospective or retrospective cohort studies with consecutive enrollment; (7) randomized controlled trials; and (8) human as a study population. Excluded criteria were: (1) the detailed brain regions and swallowing function were not described; (2) studies were not published in English.

### 2.3. Data Extraction

Two well-trained evaluators (JQ and XC) independently extracted data from the abstract, original texts, additional appendices, and protocols. Disagreements were resolved by discussion with the third author (ZMW). The studied were screened for review as per the Population, Intervention, Comparison, Outcomes, and Study design (PICOS) criteria [26]. Full texts of screened publications were examined based on the inclusion criteria and study quality. To comply with the PRISMA statement, the reviewers pilot-tested eligibility criteria and presented a flow diagram of study selection. The characteristics of study included publication year and first author, while characteristics of patients included numbers of patients, locations of stroke, phases of stroke, diagnosis methods of stroke, image analysis methods, evaluation of PSD, days to stroke evaluation, days to PSD evaluation, age, gender, rates of PSD, PSD-related brain regions, aspiration-related brain regions, and oral transit time (OTT)-related brain regions. The primary outcomes were PSD-related brain regions (including aspiration-related and OTT-related regions), and the brain regions related to dysphagia in studies were presented as percentages. The secondary outcomes were the incidence rate of PSD. If data extraction could not be completed, important missing data were first requested from the corresponding author of the studies.

### 2.4. Quality Assessment

We utilized relevant elements from the Cochrane Collaboration’s risk of bias checklist [27]. Two authors (QJ and XC) independently evaluated factors as Yes, No, or Unclear. Disagreements were resolved by discussion with the third author (ZMW). We documented additional factors, including study design, timeline for data capture, assessor-blinded, consistent assessment for all patients, declared operational definition for outcome, and outcome addressed for all patients.

### 2.5. Data Synthesis and Analysis

The brain regions with the most overlap in the included studies were considered to be most relevant to PSD. The cortical surface maps were generated by BrainNet viewer software (www.nitrc.org/projects/bnv/ (accessed on 31 October 2017) and Mricon software (www. mccauslandcenter.sc.edu/MRIcro/mricron (accessed on 2 May 2016). Differences between groups were identified by the two-tailed independent-sample *t*-tests or χ^2^ analyses for continuous variables and frequency-based variables (as appropriate). All statistical analyses were performed using SPSS software (version 23.0, SPSS/IBM, Armonk, NY, USA).

## 3. Result

A total of 3601 articles were screened, and the full text of 51 articles was reviewed. Finally, 24 studies (2306 patients) meet the inclusion criteria and are included in the final analysis. Figure 1 shows the PRISMA flow diagram. A detailed description and quality assessment of each article is provided in Table 1, Table 2, Table 3 and Table 4.

### 3.1. Characteristics of Included Studies

Of 24 studies (2306 patients), the median age of participants was 68.60 (age range from 61.1 to 75.0 years), and 57.24% (*n* = 1712) of the population were male. The median sample size was 95 (ranging from 20 to 342). Thirteen of the included studies were supratentorial strokes [11,12,13,17,25,28,29,30,31,32,33,34,35], one infratentorial stroke [36], and seven both the supratentorial and infratentorial strokes [16,37,38,39,40,41,42]. Besides, subjects of sixteen studies were in acute phase of stroke [11,12,13,14,16,17,25,29,34,35,38,39,41,42,43,44], one was in subacute phase of stroke [30], and one was in chronic phase of stroke [28], while the stroke phases of the remaining six studies were unclear (Table 1).
brainsci-12-01334-t001_Table 1Table 1The characteristics of included studies.StudyNumber of PatientsLocations of StrokePhases of StrokeDiagnosis Methods of StrokeImage Analysis MethodsEvaluation of PSDDays to Evaluation of Lesion Sites after StrokeDays to Evaluation of PSDAge (Mean ± SD, Years)Gender(M/F)Incidence of PSDPresence of Dysphagia before Stroke-EventHess 2021 [11]*n* = 132SupratentorialAcuteCTVLSMWSTNANA71.20 ± 14.2078/5463.60%NAZhang 2021 [43]*n* = 275NAAcuteMRI (DWI; DTI)VLSMWST; V-VSTWithin3 daysWithin 24 h67.92 ± 12.22182/9341.10%NoGalovic 2017 [17]*n* = 62SupratentorialAcuteMRIVLSMFOIS3 ± 2 daysWithin 48 h75.00 ± 21.0028/34NANoMoon 2018 [37]*n* = 90Supratentorial and InfratentorialNAMRIVLSMVFSSNANA68.02 ± 13.2157/33NANoMoon 2022 [36]*n* = 40CerebellarNAMRIVLSMVFSS(VDS)NANA64.02 ± 13.2124/16NANoGalovic 2013 [25]*n* = 94SupratentorialAcuteMRIROISSANAWithin 48 h74.00 ± 19.0048/4636.00%NANakamori 2021 [38]*n* = 342Supratentorial and InfratentorialAcuteMRI (FLAIR)NAVFSSWithin 1 weekWithin 14 days70.40 ± 12.60200/14213.20%NAJang 2017 [28]*n* = 82SupratentorialChronicMRIVLSMVFSSNANA73.90 ± 8.0175/773.17%NoLapa 2021 [29]*n* = 113SupratentorialAcuteCT or MRIASPECTSFEES(FEDSS)NAWithin 24 h69.00 ± 13.0067/4554.90%NoWilmskoetter 2019 [13]*n* = 68SupratentorialAcuteDWIVLSMMBSImP; PASNANA68.21 ± 15.2332/36NANoSuntrup 2015 [44]*n* = 200NAAcuteCT or MRINAFEES(FEDSS)Within 24–60 h Within 96 h73.70 ± 16.50101/9982.50%NoFlowers 2017 [14]*n* = 160NAAcuteMRINANAWithin 14 daysNA68.0091/6948.00%NAKim 2016 [30]*n =* 31SupratentorialSubacuteDTIFA value; ADC valueVFSS(VDS)NANA61.10 ± 9.4219/1254.80%NoIm 2018 [31]*n* = 21SupratentorialNAMRINAVFSSNAWithin 14 days57.38 ± 12.7113/8NANoOsawa 2013 [39]*n* = 50Supratentorial and InfratentorialAcuteCT or MRISPECT dataVFSS; RSST; MWSTNANA70.20 ± 10.3032/1870.00%NoMomosaki 2012 [32]*n* = 20SupratentorialNAMRIrCBFMWST; FEESFOISNAWithin 7 days66.10 ± 5.1014/6NANoCola 2010 [12]*n* = 20SupratentorialAcuteMRINAVFSSNANA62.30 ± 12.2019/135.00%NoSaito 2016 [40]*n* = 20Supratentorial and InfratentorialNAMRI (DWI; FLAIR)NAVFSSNAWithin 4 weeks76.40 ± 10.407/13NANADehaghani 2016 [16]*n* = 113Supratentorial and InfratentorialAcuteCT or MRINAMASAWithin 24–72 hWithin 20 days64.37 ± 15.1069/4447.80%NoDaniels 1996 [24]*n* = 16SupratentorialNACT or MRINAVFSSwithin 1 monthwithin 1 month66.60 ± 13.9012/4NANABroadley 2003 [41]*n* = 149Supratentorial and InfratentorialAcuteCT or MRINAParramatta Hospitals AssessmentNAWithin 72 h72.0088/6150.00%NASteinhagen 2009 [42]*n* = 60Supratentorial and InfratentorialAcuteCT or MRINAFEESNANA74.60 ± 11.4025/35NANoGonzalez-Fernandez 2008 [34]*n* = 14SupratentorialAcuteMRI (FLAIR;DWI)ROIsNAWithin 24 hWithin 7 days62.60 ± 14.307/7NANoGalovic 2016 [35]*n* = 119SupratentorialAcuteMRI (DWI)VLSMBogenhausen Dysphagia Score part 2NAWithin 48 h76.00 ± 9.0065/54NANoNote: CT, Computed Tomography; MRI, Magnetic Resonance Imaging; FLAIR, Fluid Attenuated Inversion Recovery; DWI, Diffusion-Weighted Imaging; VLSM, Voxel-based Lesion Symptom Mapping; ROI, Region of Interest; FA value, Fractional Anisotropy value; ADC value, Apparent Diffusion Coefficient value; SPECT, Single-photon Emission Computed Tomography; rCBF, Regional Cerebral Blood Flow; WST, Water Swallowing test; V-VST, Volume Viscosity Screening Test; VFSS, Videofluoroscopic Swallowing Study; FOIS, Functional Oral Intake Scale; PAS, Penetration-aspiration Scale; VDS, Videofuoroscopic Dysphagia Scale; FEES, Fiberoptic Endoscopic Evaluation of Swallowing; FEDSS, Fiberoptic Endoscopic Dysphagia Severity Scale; RSST, Repetitive Saliva Swallowing Test; MWST, Modified Water Swallow Test; SSA, Standardized Swallowing Assessment; NA, not applicable.

### 3.2. Study Design and Quality Assessment

As shown in Table 2, five of the included articles were prospective studies [25,34,38,41,42], five articles were retrospective ones [13,24,30,39,40], while the types of other fourteen articles were unclear. A total of fourteen articles reported being assessor-blinded [11,12,13,17,28,29,30,34,38,40,41,42,43,44], and the other ten were unclear. Twenty-three articles reported consistent assessment for all patients, and the remaining one was unclear [25]. Eighteen articles reported declared operational definition for the outcome, and the remaining six were unclear [13,14,25,34,35,41]. Twenty-three articles reported outcomes addressed for all patients, and the remaining one did not report [41].
brainsci-12-01334-t002_Table 2Table 2Evaluation of study quality.StudyTimeline for Data CaptureAssessor BlindedConsistent Assessment for All PatientsDeclared Operational Definition for OutcomeOutcome Addressed for All PatientsHess 2021 [11]unclearyesyesyesyesZhang 2021 [43]unclearyesyesyesyesGalovic 2017 [17]unclearyesyesyesyesMoon 2018 [37]unclearunclearyesyesyesMoon 2022 [36]unclearunclearyesyesyesGalovic 2013 [25]prospectiveunclearunclearunclearyesNakamori 2021 [38]prospectiveyesyesyesyesJang 2017 [28]unclearyesyesyesyesLapa 2021 [29]unclearyesyesyesyesWilmskoetter 2019 [13]retrospectiveyesyesunclearyesSuntrup 2015 [44]unclearyesyesyesyesFlowers 2017 [14]unclearunclearyesunclearyesKim 2016 [30]retrospectiveyesyesyesyesIm 2018 [31]unclearunclearyesyesyesOsawa 2013 [39]retrospectiveunclearyesyesyesMomosaki 2012 [32]unclearunclearyesyesyesCalo 2010 [12]unclearyesyesyesyesSaito 2016 [40]retrospectiveyesyesyesyesDehaghani 2016 [36]unclearunclearyesyesyesDaniels 1996 [24]retrospectiveunclearyesyesyesBroadley 2003 [41]prospectiveyesyesunclearnoSteinhagen 2009 [42]prospectiveyesyesyesyesGonzalez-Fernandez 2008 [34]prospectiveyesyesunclearyesGalovic 2016 [35]unclearunclearyesunclearyes

### 3.3. Swallowing Assessment

The swallowing assessment tools included the Water Swallowing Test (WST) [11,43], Videofluoroscopic Swallowing Study (VFSS) (taking up 37.5% of included studies) [12,24,28,30,31,36,38,39,40], Volume-viscosity Swallow Test (V-VST) (4.2%) [43], Fiberoptic Endoscopic Evaluation of Swallowing (FEES) (12.5%) [29,42,44], MBS impairment tool (MBSImP) (4.2%) [13], Standardized Clinical Assessment Tool (4.2%) [25], and Bogenhausen Dysphagia Score Part 2 (BDS-2) (4.2%) [35]. A total of thirteen articles described the days to the evaluation of PSD (ranging from 24 h to 4 weeks; Table 1).

### 3.4. Diagnosis of Stroke

The MRI was used for the diagnosis of stroke in the 24 articles, in which four articles reported using DWI scans [13,34,35,43], two articles adopted DTI scans [30,43], three articles used flair scans [34,38,40], and the remaining twelve articles did not report the scanning sequence. Of all the articles, eight articles adopted VLSM in the assessment of stroke-related brain regions [11,13,17,28,35,36,37,43], and eight described the days to evaluation of lesion sites after stroke (ranging from 24 h to 14 days; Table 1) [14,16,17,24,34,38,43,44].

### 3.5. PSD-Related Lobar and Deep Brain Regions

The PSD-related lobar and deep brain regions included insular cortex (including the right, left, and anterior insular cortex) [11,13,14,16,17,24,25,29,31,35,37,39,41], which has been reported in 54.2% of included studies; frontal lobe (16.7%) (including superior frontal gyrus, inferior frontal gyrus, left inferior frontal lobe, right inferior frontal gyrus, middle frontal gyrus) [3,28,37,40]; temporal gyrus (4.2%) [11]; left and right basal ganglia (25%) (including right internal capsule, bilateral posterior limb of the internal capsule, lentiform nucleus) [11,28,29,34,37,38]; corona radiata (16.7%) (including left corona radiata and superior corona radiata) [13,16,25,34]; postcentral (12.5%) (including right primary sensory cortex) [13,16,44]; precentral (16.7%) (including left primary motor cortex, motor supplementary areas) [13,28,30,44]; precuneus (4.2%) [39] (Table 3 and Figure 2).brainsci-12-01334-t003_Table 3Table 3The PSD-related brain regions, aspiration-related brain regions, and OTT-related brain regions.StudyNumber of PatientsPSD-Related Brain RegionsAspiration-Related Brain RegionsOTT-Related Brain RegionsHess 2021 [11]*n* = 132Right insular cortex; Left basal ganglia; Left corona radiata; Left central regionNANAZhang 2021 [43]*n* = 275Left inferior parietal gyrusNANAGalovic 2017 [17]*n* = 62Superior corona radiata; Anterior insular cortexNANAMoon 2018 [37]*n* = 90Superior frontal gyrus; Inferior frontal gyrus; Lentiform nucleus; Insular cortexNALentiform nucleus; Insular cortexMoon 2022 [36]*n* = 40Posterior lobe of the left cerebellumNANAGalovic 2013 [25]*n* = 94Internal capsule; Insular cortexInsular cortexNANakamori 2021 [38]*n* = 342Parietal lobe lesion; Basal gangliaParietal lobeNAJang 2017 [28]*n* = 82Left inferior frontal lobe; Precentral gyrus; Right basal ganglia; Corona radiate; PutamenPutamenPrecentral gyrusLapa 2021 [29]*n* = 113Left lentiform nucleus; Left insular cortex; Left frontal operculum; Right insular cortexNANAWilmskoetter 2019 [13]*n* = 68Right inferior frontal gyrus; Pre- and postcentral gyrus; Supramarginal gyrus; Angular gyrus; Superior temporal gyrus; Insular cortex; Thalamus; Amygdala; Superior longitudinal fasciculus; Corona radiata; Internal capsule; External capsule; Ansalenticularis; Lenticular fasciculusNANASuntrup 2015 [44]*n* = 200Right pre- and post-central gyri; Opercular region; Supramarginal gyrus; Respective subcortical white matter tracts; Post-central lesionNANAFlowers 2017 [14]*n* = 160Medullary; Insular cortex; PontineNANAKim 2016 [30]*n* = 31Primary motor cortex on the contra-lesional side; Bilateral posterior limb of the internal capsuleNANAIm 2018 [31]*n* = 21Caudate nucleus; Insular cortexCaudate nucleusNAOsawa 2013 [39]*n* = 50Left precuneus; Left insular cortex; Anterior cingulate gyrusAnterior cingulate gyrusNAMomosaki 2012 [32]*n* = 20Brodmann area 4NANACalo 2010 [12]*n* = 20Left periventricular white matterNANASaito 2016 [40]*n* = 20Middle frontal gyrusNANADehaghani 2016 [16]*n* = 133Right primary sensory; Right insular cortex; Right internal capsuleNANADaniels 1996 [24]*n* = 16Insular cortexNANABroadley 2003 [41]*n* = 149Frontal cortex; Insular cortexNANASteinhagen 2009 [42]*n* = 60NAInsular cortexNAGonzalez-Fernandez 2008 [34]*n* = 14Primary somatosensory; Motor and motor supplementary areas; Putamen; Caudate; Basal ganglia; Internal capsule; NANAGalovic 2016 [35]*n* = 119Anterior insular cortexNANANote: PSD, post-stroke dysphagia; OTT, oral transit time; NA, not applicable.

A total of twelve studies reported the effect of the insular cortex on PSD. These studies were conducted on an acute phase of stroke, investigated the potential lesion pattern related to PSD, and found that the right insular cortex is related to swallowing dysfunction and predictive for the development of dysphagia [11,16,29]. According to Hess et al., the MNI coordinates were X = −39, Y = −11, Z = 10, and the voxels were 799 [11]. Another two studies, conducted on right hemispheric strokes, reported that associations were found in the left insular cortex [29,39]. Furthermore, two studies conducted on supratentorial strokes by VLSM analysis demonstrated that the anterior insular cortex was associated with the prognosis of PSD [17,35]. The anterior insular cortex (MNI coordinates were X = 39, Y = 10, Z = 20) was related to impaired oral intake 4 weeks after stroke, as reported by Galovic et al. [17], and affected 54% of voxels. Besides, Galovic et al. reported the anterior insular cortex was also related to the time before oral feeding [35] and found a significant difference in the anterior insular cortex (MNI coordinates are X = 37, Y = 10, Z = 6) by the comparison between tube-dependency and no tube feeding patients, which affected 70% of voxels. The remaining six studies did not report the specific regions of the insular cortex.

### 3.6. Aspiration-Related and Oral Transit Time (OTT)-Related Brain Regions

The aspiration-related brain regions included insular cortex (8.3%) [25,42], parietal lobe (4.17%) [38], putamen (4.17%) [28], caudate nucleus (4.17%) [31], and anterior cingulate gyrus (4.17%) [39]. The OTT-related brain regions included insular cortex, lentiform nucleus, and precentral gyrus [28,37].

### 3.7. The Incidence Rate of PSD

The incidence rates of PSD were 63.6%, 41.1%, 54.9%, 82.5%, 48%, 35%, 47.8%, and 50% in acute phase of stroke [11,12,14,16,29,41,43,44], while the rate was 54.8% in subacute phase of stroke [29]. For supratentorial stroke patients, the reported incidence rates of PSD were 54.9%, 54.8%, and 35% [12,29,30]. Furthermore, the incidence rate of aspiration was 36%, 13.2%, and 70.0% in acute phase of stroke [25,38,39].

We conducted a secondary analysis according to the data provided in the included articles. The results showed that the incidence rate of PSD was around 40.4%, which was significantly higher in the male than in the female population (χ^2^ = 196.17, *p* < 0.001), while there was no statistical difference in incidence rate between ischemic and hemorrhagic stroke groups (χ^2^ = 1.173, *p* = 0.279), as well as right and left hemispheric stroke groups (χ^2^ = 0.648, *p* = 0.412) (Table 4).brainsci-12-01334-t004_Table 4Table 4The incidence of PSD stratified by gender, type of stroke, and location of stroke.StudyNumber of PatientsPSD vs. No PSDMale vs. FemaleIschemic vs. HemorrhagicRight vs. LeftInfratentorial vs. SupratentorialHess 2021 [11]*n* = 13284/4848/36NA36/4810/74Zhang 2021 [43]*n* = 275113/16275/38NA52/43NAGalovic 2017 [17]*n* = 62NANANANANAMoon 2018 [37]*n* = 9090/057/3364/2650/3516/74Moon 2022 [36]*n* = 40NANANANANAGalovic 2013 [25]*n* = 94NANANANANANakamori 2021 [38]*n* = 34245/29737/18NANANAJang 2017 [28]*n* = 8282/075/768/1426/11NALapa 2021 [29]*n* = 11362/5127/24NA11/40NAWilmskoetter 2019 [13]*n* = 68NA32/36NANANASuntrup 2015 [44]*n* = 200NANANANANAFlowers 2017 [14]*n* = 16076/8446/30NANANAKim 2016 [30]*n* = 3117/1412/5NA13/14NAIm 2018 [31]*n* = 2121/013/814/79/12NAOsawa 2013 [39]*n* = 5027/2313/1419/84/1213/3Momosaki 2012 [32]*n* = 2010/108/22/8NANACalo 2010 [12]*n* = 207/14NANA10/10NASaito 2016 [40]*n* = 2020/07/13NA8/12NADehaghani 2016 [36]*n* = 13354/7924/308/12NANADaniels 1996 [24]*n* = 16NANANA8/8NABroadley 2003 [41]*n* = 14974/7541/33NANANASteinhagen 2009 [42]*n* = 60NANANANA16/44Gonzalez-Fernandez 2008 [34]*n* = 2914/157/7NA10/4NAGalovic 2016 [35]*n* = 11912/1076/6NA5/7NANote: PSD, post-stroke dysphagia. NA, not applicable.

## 4. Discussion

The present study found that the PSD-related lobar and deep brain regions included the insular cortex, frontal lobe, temporal gyrus, basal ganglia, postcentral, precentral, precuneus, corona radiate, etc., in which the insular cortex might be most relevant to PSD and aspiration after PSD and was reported in 54.2% of included studies. Furthermore, the total incidence rate of PSD was around 40.4%, and the incidence of male was nearly 2.57 times as much as that of female.

### 4.1. The Lobar and Deep Brain Regions Participate in the Swallowing Function Regulation

The swallowing function is not only regulated by the medulla oblongata, but by the lobar and deep brain regions. Different lobar and deep brain regions participate in different aspects of swallowing function. For example, lobar regions like the parietal–temporal lobes are associated with oropharyngeal residue, while the somatosensory cortex governs and executes motions by controlling and providing feedback to the brainstem and is responsible for the laryngeal elevation and vestibular closure [13,45]. Besides, the deep brain regions are also involved in PSD. The basal ganglia are considered to participate in the sensory input of swallowing function [34,46], in which the internal capsule is involved in the oropharyngeal residue and aspiration after dysphagia [13,34], while the outer capsule is involved in laryngeal elevation and vestibular closure [13,35]. The periventricular white matter is related to the occurrence of PSD [12,31], and the corona radiata is related to the oropharyngeal residue, laryngeal elevation, and vestibular closure [13].

### 4.2. Insular Cortex May Be Most Relevant to PSD

The present study found that the insular cortex may be most relevant to PSD. The insular cortex is involved in an overwhelming variety of functions, including decision-making, complex social functions, addiction, and sensory processing, to represent feelings [47,48]. It is located in the deep brain part of the lateral fissure and covered by the parietal, frontal, and temporal lobes, which accept the projection fiber from the thalamic nucleus and participate in the swallowing coordination by sensory-motor integration [13]. Therefore, the insular cortex participates in the various aspects of the swallowing process, including OTT, tube dependency, pharyngeal transit time, and aspiration [17,25,31,35,37]. Damage to the insular cortex (e.g., brain trauma) is more likely to manifest as delayed swallowing initiation, decreased laryngeal elevation, and impairment of laryngeal vestibular closure [13,49,50].

### 4.3. Insular Cortex May Be Relevant to Aspiration after PSD

The aspiration is a common but serious sequela after PSD and is associated with abnormal swallow–breathing coordination [51]. The insular cortex may be involved in the aspiration process by participating in the swallow–breathing coordination. The swallow–breathing coordination center is often considered the medulla oblongata, in which the swCPG and respiratory center pattern generator (rCPG) participates in the regulation of swallow–breathing coordination directly [9,10]. A core aspect of swallow–breathing coordination is the reciprocal inhibition between swCPG and rCPG, and any injury or damage to the brainstem may lead to PSD and aspiration [52].

Meanwhile, the lobar and deep brain neural networks also play an important role in swallow–breathing coordination. On the one hand, the insular cortex participates in the coordination of swallowing function. The previous studies adopted fMRI to explore the features of the cerebral cortex for PSD patients and found that the insular cortex was activated obviously during the swallowing task [50]. For healthy volunteers, the insular cortex was also activated, and the functional connection was enhanced between the insular cortex and other brain regions during swallowing tasks, including the sensorimotor cortex, frontal lobe, and parietal lobe [49]. Besides, damage to the anterior insular would cause more serious symptoms, manifesting as severely impaired oral intake requiring acute tube insertion [35].

On the other hand, the insular cortex also participates in the coordination of respiratory function. Brain imaging studies have provided evidence that dyspnea is associated with activation of the insular cortex [53]. Meanwhile, Trevizan-Baú et al. adopted holera toxin subunit B (CT-B) for the retrograde tracing of the neural regulation of breathing and found that insular exists alongside a great number of neurons with CT-B labeled [54]. Van et al. used the pseudorabies virus (PRV) inoculation into the thyroarytenoid muscle, which participates in the breathing–swallowing coordination, showing that the PRV transfer from the peripheral to the swCPG, rCPG, hypothalamus, insular, and motor cortex. Besides, the different insular cortex regions might be involved in different breathing patterns, and damage to the posterior insular cortex is more likely to manifest as respiratory excitatory responses, while the anterior insular manifests as inhibitory respiratory responses [55]. Therefore, the insular cortex (especially the anterior insular) might be involved in the occurrence of aspiration after PSD by participating in the regulation of swallow–breathing coordination.

According to the previous research, we proposed a new hypothesis for the mechanism of the insular cortex on aspiration after PSD. Firstly, the brainstem receives input signals from peripheral organs (including the tongue, bronchial, and esophagus), in which the swCPG and rCPG located in the brainstem are reciprocal inhibition. Secondly, the thalamus receives input signals from the brainstem. Eventually, the insular cortex receives input signals from the thalamus. The stroke in the insular cortex may disturb the reciprocal inhibition relationship of swCPG and rCPG, which may lead to PSD and aspiration after PSD (Figure 3).

### 4.4. Clinical/Rehabilitative Implication of Normal Function for Insular Cortex

Several suggestions can be recommended according to our results. Firstly, the insular cortex participates in the various aspects of swallowing and can be a potentially promising target for the treatment of aspiration. For example, noninvasive brain stimulation (NIBS), including repetitive transcranial magnetic stimulation (rTMS) and transcranial direct current stimulation (tDCS), is a practical technique which has proved effective, and is widely used to promote the recovery of PSD [56]. However, the rTMS relies on accurate stimulation at specific brain regions to achieve clinical efficacy [57]. Therefore, the insular cortex may become the target in the treatment of PSD and aspiration. Secondly, brainstem stroke is reportedly the main incentive of PSD [58], however, the lobar and deep brain regions also participate in the swallowing function regulation [59]. After lobar and deep brain regions stroke, a comprehensive assessment may be needed to avoid serious complications according to our results.

### 4.5. Limitations

The present study has several limitations. First, the relationship between the brain stem and dysphagia is not investigated in the present study. Second, only studies that presented detailed stroke lesion sites were included, while those on the large areas of the brain regions were excluded, which might cause potential bias. Third, the studies included in the present research were limited to those in English, which may lead to bias. Fourth, the relationship between brain lesions and dysphagia after stroke was investigated based on qualitative analysis rather than quantitative analysis due to the limited data. Therefore, high-quality quantitative analysis studies are needed.

### 4.6. Conclusions

The PSD-related lobar and deep brain regions included the insular cortex, frontal lobe, parietal lobe, basal ganglia, etc., in which insular cortex may be the area most relevant to PSD and aspiration after PSD.

## Figures and Tables

**Figure 1 brainsci-12-01334-f001:**
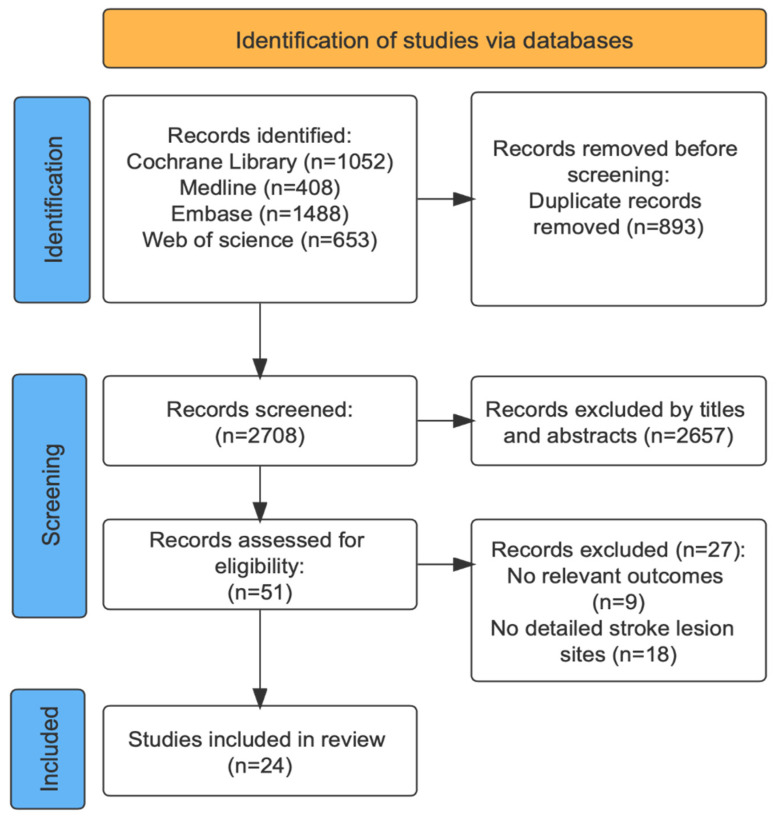
The PRISMA flow diagram.

**Figure 2 brainsci-12-01334-f002:**
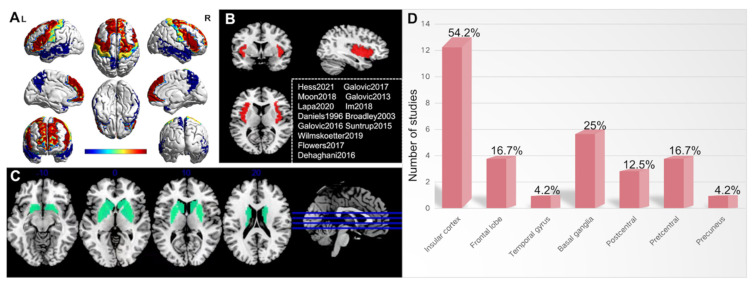
Illustration of lesion overlap and distribution for patients with post-stroke dysphagia (PSD). The insular cortex might be the most relevant brain region for PSD. (**A**) Lobar brain regions related to PSD. (**B**) Insular cortex and corresponding published articles in this review. (**C**) Deep brain regions related to PSD (including basal ganglia). (**D**) The percentages of specific brain regions related to PSD in the included studies. Note: the detailed information (including the lesion reported and the related studies) was shown in Table 2; L, left; R, right.

**Figure 3 brainsci-12-01334-f003:**
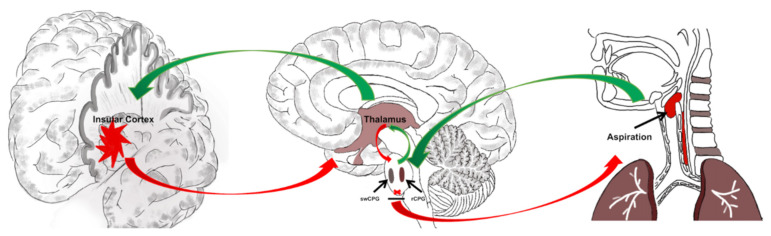
A hypothesis for the mechanism of the insular cortex on aspiration after PSD based on previous studies. Firstly, the brainstem receives input signals from peripheral organs (including the tongue, bronchial, and esophagus), in which the swCPG and rCPG located in the brainstem are reciprocal inhibition. Secondly, the thalamus receives input signals from the brainstem. Eventually, the insular cortex receives input signals from the thalamus. The stroke in the insular cortex may disturb the reciprocal inhibition relationship of swCPG and rCPG, which may lead to PSD and aspiration after PSD. Note: PSD, post-stroke dysphagia; swCPG, swallowing center pattern generator; rCPG, respiratory center pattern generator. Green arrow, promotion; red arrow, inhabitation.

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
