# Peer review of "Effects of Insular Cortex on Post-Stroke Dysphagia: A Systematic Review and Meta Analysis"

_brainsci, 2022, doi:10.3390/brainsci12101334_

Round 1

Reviewer 1 Report

Stroke is the main cause of mortality in developed countries, they should update their bibliography.

In the selection criteria, they only selected scientific publications in English, which gives rise to a very important publication bias since their results cannot be extrapolated. Why have they done that?

Previous reviews should not be included, but they should be searched in case any important study has escaped us, so it is not an exclusion criterion.

Tables 1,2,3 and 4 should be improved, the years of publication give rise to errors and make reading difficult. Also, the margins are not justified. The N is not capitalized, it is lowercase and italicized. 

From line 226 the letter changes size

Author Response

  1. Stroke is the main cause of mortality in developed countries, they should update their bibliography.

Authors’ response: Thanks for your suggestion. We revised the manuscript reference to the suggestions, and all changes have been marked in red.

  1. In the selection criteria, they only selected scientific publications in English, which gives rise to a very important publication bias since their results cannot be extrapolated. Why have they done that?

Authors’ response: Thanks for your suggestion. We selected scientific publications in English, which is more easily recognized and understand. However, it may lead to important publication bias, this has been listed in the limitation part in the present study (on Page 12).

  1. Previous reviews should not be included, but they should be searched in case any important study has escaped us, so it is not an exclusion criterion.

Authors’ response: Thanks for your suggestion. We revised the exclusion criterion according to the suggestions (on Page 5). During the screening and writing step, the previous reviews have been referenced by authors in the present study to avoid missing important study.

  1. Tables 1,2,3 and 4 should be improved, the years of publication give rise to errors and make reading difficult. Also, the margins are not justified. The N is not capitalized, it is lowercase and italicized. 

Authors’ response: Thanks for your suggestion. We have revised the Table1-4. The margins are justified, and the N is lowercase and italicized.

  1. From line 226 the letter changes size

Authors’ response: Sorry about the mistake, we have revised the manuscript.

Reviewer 2 Report

Dear Authors,

thank you for giving me the opportunity to revise your manuscript. This  review is focused on the investigation of brain regions involved and affected in post-stroke dysfagia (PSD). The paper is well written, but there are some critical issues to be addressed:

Introduction: Dysfagia in post-stroke patient in a multifactorial impairment, involving different mechanism, both central and peripheral. The buccal coordination plays a central role in dysfagia in this patient, such as other factors aging-related. I suggest to improve the introduction in this direction, exploring the different causes of dysfagia in a global vision. Please, read "de Sire A, Ferrillo M, Lippi L, Agostini F, de Sire R, Ferrara PE, Raguso G, Riso S, Roccuzzo A, Ronconi G, Invernizzi M, Migliario M. Sarcopenic Dysphagia, Malnutrition, and Oral Frailty in Elderly: A Comprehensive Review. Nutrients. 2022 Feb 25;14(5):982. doi: 10.3390/nu14050982. and " de Sire A, Baricich A, Ferrillo M, Migliario M, Cisari C, Invernizzi M. Buccal hemineglect: is it useful to evaluate the differences between the two halves of the oral cavity for the multidisciplinary rehabilitative management of right brain stroke survivors? A cross-sectional study. Top Stroke Rehabil. 2020 Apr;27(3):208-214. doi: 10.1080/10749357.2019.1673592."

Methods:

1) Please, add an explicative table to describe the MASH terms for each database. 

2) It should be desirable to add the PICO model in the methods section

3) Regarding swallowing assessment, it should be adequate to specify which study have used the different methods

4) In table 1 the authors affirm to have not data about the location of stroke in several included studies, but all patients undergone TC/MRI to assess the different lobar area in PSD. This point is not well clear, and I found strange this information. Why is there this gap?

5) I suggest to put the reference for each brain region or percentage, to make clearer the reading 

Results:

1) The authors describe the different lobar point involved in PSD. This paragraph is a cornerstone of the paper, but its explication is poor. They should list the different results of the study in more detailed manner

2) It would be desirable to specify the presence of dysfagia before stroke-event in each study, if possible.  

Discussion:

1) Please, change the format from line 224 to 262.

2) The discussion in lack of information and i suggest to rewrite it at the light of the heterogeneity of  results. For example, the authors underline the presence of different lobar area involved in PSD, but focus their attention only on the insular cortex. Please, stress this point.

3) Which are the clinical/rehabilitative implication of your work? please, add a paragraph to stress this point

A moderate english revision is requested

Best Regards

Author Response

Reviewer 2

thank you for giving me the opportunity to revise your manuscript. This  review is focused on the investigation of brain regions involved and affected in post-stroke dysfagia (PSD). The paper is well written, but there are some critical issues to be addressed.

Introduction:

Dysfagia in post-stroke patient in a multifactorial impairment, involving different mechanism, both central and peripheral. The buccal coordination plays a central role in dysfagia in this patient, such as other factors aging-related. I suggest to improve the introduction in this direction, exploring the different causes of dysfagia in a global vision. Please, read "de Sire A, Ferrillo M, Lippi L, Agostini F, de Sire R, Ferrara PE, Raguso G, Riso S, Roccuzzo A, Ronconi G, Invernizzi M, Migliario M. Sarcopenic Dysphagia, Malnutrition, and Oral Frailty in Elderly: A Comprehensive Review. Nutrients. 2022 Feb 25;14(5):982. doi: 10.3390/nu14050982. and " de Sire A, Baricich A, Ferrillo M, Migliario M, Cisari C, Invernizzi M. Buccal hemineglect: is it useful to evaluate the differences between the two halves of the oral cavity for the multidisciplinary rehabilitative management of right brain stroke survivors? A cross-sectional study. Top Stroke Rehabil. 2020 Apr;27(3):208-214. doi: 10.1080/10749357.2019.1673592."

Authors’ response: Thanks for your suggestion. We revised the manuscript reference to the suggestions as follows (on Page 3): Besides, other PSD-related factors include advanced age, and unilateral spatial neglect (USN), etc. because of the decline of physiological functions.[1-2]

Reference

[1] De Sire A, Ferrillo M, Lippi L, Agostini F, de Sire R, Ferrara PE, Raguso G, Riso S, Roccuzzo A, Ronconi G, Invernizzi M, Migliario M. Sarcopenic Dysphagia, Malnutrition, and Oral Frailty in Elderly: A Comprehensive Review. Nutrients. 2022, 14(5):982.

[2] De Sire A, Baricich A, Ferrillo M, Migliario M, Cisari C, Invernizzi M. Buccal hemineglect: is it useful to evaluate the differences between the two halves of the oral cavity for the multidisciplinary rehabilitative management of right brain stroke survivors? A cross-sectional study. Top Stroke Rehabil. 2020, 27(3):208-214.

Methods: 

  • Please, add an explicative table to describe the MASH terms for each database. 

Authors’ response: Thanks for your suggestion. We have uploaded an explicative table to describe the MASH terms for each database in the Supplemental Appendix. The table is as follow:

Database

Mesh terms

Medline

Stroke: Strokes; Cerebrovascular Accident; Cerebrovascular Accidents; CVA (Cerebrovascular Accident); CVAs (Cerebrovascular Accident); Cerebrovascular Apoplexy; Apoplexy, Cerebrovascular; Vascular Accident, Brain; Brain Vascular Accident; Brain Vascular Accidents; Vascular Accidents, Brain; Cerebrovascular Stroke; Cerebrovascular Strokes; Stroke, Cerebrovascular; Strokes, Cerebrovascular; Apoplexy; Cerebral Stroke; Cerebral Strokes; Stroke, Cerebral; Strokes, Cerebral; Stroke, Acute; Acute Stroke; Acute Strokes; Strokes, Acute; Cerebrovascular Accident, Acute; Acute Cerebrovascular Accident; Acute Cerebrovascular Accidents; Cerebrovascular Accidents, Acute.

Dysphagia: Deglutition Disorder; Disorders, Deglutition; Swallowing Disorders; Swallowing Disorder; Oropharyngeal Dysphagia; Dysphagia, Oropharyngeal; Esophageal Dysphagia; Dysphagia, Esophageal.

Cochrane library

Stroke: Cerebral Strokes; CVAs (Cerebrovascular Accident); Cerebral Stroke; Strokes, Cerebral; Cerebrovascular Stroke; Strokes, Cerebrovascular; Apoplexy; Stroke, Cerebral; Cerebrovascular Accident; Strokes; Vascular Accidents, Brain; Vascular Accident, Brain; Stroke, Cerebrovascular; Cerebrovascular Strokes; CVA (Cerebrovascular Accident); Brain Vascular Accident; Apoplexy, Cerebrovascular; Brain Vascular Accidents; Cerebrovascular Apoplexy; Cerebrovascular Accidents; Cerebrovascular Accidents, Acute; Strokes, Acute; Acute Stroke; Acute Cerebrovascular Accidents; Stroke, Acute; Cerebrovascular Accident, Acute; Acute Cerebrovascular Accident; Acute Strokes.

Dysphagia: Dysphagia; Oropharyngeal; Oropharyngeal Dysphagia; Swallowing Disorders; Swallowing Disorder; Disorders, Deglutition; Dysphagia; Deglutition Disorder; Dysphagia, Esophageal; Esophageal Dysphagia.

Web of science

Stroke: Strokes; Cerebrovascular Accident; Cerebrovascular Accidents; CVA (Cerebrovascular Accident); CVAs (Cerebrovascular Accident); Cerebrovascular Apoplexy; Apoplexy, Cerebrovascular; Vascular Accident, Brain; Brain Vascular Accident; Brain Vascular Accidents; Vascular Accidents, Brain; Cerebrovascular Stroke; Cerebrovascular Strokes; Stroke, Cerebrovascular; Strokes, Cerebrovascular; Apoplexy; Cerebral Stroke; Cerebral Strokes; Stroke, Cerebral; Strokes, Cerebral; Stroke, Acute; Acute Stroke; Acute Strokes; Strokes, Acute; Cerebrovascular Accident, Acute; Acute Cerebrovascular Accident; Acute Cerebrovascular Accidents; Cerebrovascular Accidents, Acute.

Dysphagia: Deglutition Disorder; Disorders, Deglutition; Swallowing Disorders; Swallowing Disorder; Oropharyngeal Dysphagia; Dysphagia, Oropharyngeal; Esophageal Dysphagia; Dysphagia, Esophageal.

Embase

Stroke: accident, cerebrovascular; acute cerebrovascular lesion; acute focal cerebral vasculopathy; acute stroke; apoplectic stroke; apoplexia; apoplexy; blood flow disturbance, brain; brain accident; brain attack; brain blood flow disturbance; brain insult; brain insultus; brain vascular accident; cerebral apoplexia; cerebral insult; cerebral stroke; cerebral vascular accident; cerebral vascular insufficiency; cerebrovascular accident; cerebrovascular arrest; cerebrovascular failure; cerebrovascular injury; cerebrovascular insufficiency; cerebrovascular insult; cerebrum vascular accident; cryptogenic stroke; CVA; insultus cerebralis; ischaemic seizure; ischemic seizure; stroke; thrombotic stroke.

Dysphagia: aphagopraxia; deglutition difficulty; deglutition disorder; deglutition disorders; difficult deglutition; difficulty in swallowing; difficulty swallowing; dysphagias; swallowing difficult; swallowing difficultness; swallowing difficulty; swallowing disorder.

  • It should be desirable to add the PICO model in the methods section

Authors’ response: Thanks for your suggestion. We revised the manuscript reference to the suggestions as follows (on Page 5): The studied were screened for review as per the Population, Intervention, Comparison, Outcomes and Study design (PICOS) criteria[1].

Reference:

[1] Khan QI, Baig H, Al Failakawi A, Majeed S, Khan M, Lucocq J. The Effect of Platelet-Rich Plasma on Healing Time in Patients Following Pilonidal Sinus Surgery: A Systematic Review. Cureus. 2022 Aug 8;14(8):e27777.

  • Regarding swallowing assessment, it should be adequate to specify which study have used the different methods.

Authors’ response: Thanks for your suggestion. We have revised the manuscript, and added the reference for each swallowing assessment method (on Page 9). Revised as follows: The swallowing assessment tools included the Water Swallowing Test (WST)11,43, Videofluoroscopic Swallowing Study (VFSS) (taking up 37.5% of included studies),12,24,28,30,31,36,38-40 Volume-viscosity Swallow Test (V-VST) (4.2%),43 Fiberoptic Endoscopic Evaluation of Swallowing (FEES) (12.5%),29,42,44 MBS impairment tool (MBSImP) (4.2%),13 Standardized Clinical Assessment Tool (4.2%),25 and Bogenhausen Dysphagia Score Part 2 (BDS-2) (4.2%).35 A total of thirteen articles described the days to the evaluation of PSD (ranging from 24h to 4 weeks; Table 1). More detail were presented in Table 1.

  • In table 1 the authors affirm to have not data about the location of stroke in several included studies, but all patients undergone TC/MRI to assess the different lobar area in PSD. This point is not well clear, and I found strange this information. Why is there this gap?

Authors’ response: Sorry about we didn’t clarify this in Table 1. The authors recheck the included studies, and Zhang2021, Suntrup2015, Flowers2017 have not data about the location of study, even though CT or MRI was used to confirm stroke.

5) I suggest to put the reference for each brain region or percentage, to make clearer the reading 

Authors’ response: Thanks for your suggestion. We revised the manuscript including Abstract, Method, and Result part reference to the suggestions, and each brain region has been described as percentage. 

Results: 

  • The authors describe the different lobar point involved in PSD. This paragraph is a cornerstone of the paper, but its explication is poor. They should list the different results of the study in more detailed manner

Authors’ response: Sorry about the poor explication. We have revised the PSD-related lobar and deep brain regions part (on Page 8). Revised as follows: The PSD-related lobar and deep brain regions included insular cortex (including the right, left, and anterior insular cortex),11,13,14,16,17,24,25,29,31,35,37,39,41 which has been reported in 54.2% of included studies; frontal lobe (16.7%) (including superior frontal gyrus, inferior frontal gyrus, left inferior frontal lobe, right inferior frontal gyrus, middle frontal gyrus);3,28,37,40 temporal gyrus (4.2%);11 left and right basal ganglia (25%) (including right internal capsule, bilateral posterior limb of the internal capsule, lentiform nucleus);11,28,29,34,37,38 corona radiata (16.7%) (including left corona radiata and superior corona radiata);13,16,25,34 postcentral (12.5%) (including right primary sensory cortex);13,16,44 pretcentral (16.7%) (including left primary motor cortex, motor supplementary areas);13,28,30,44 precuneus (4.2%);39 and periventricular white matter (4.2%).12 (Table 2 and Fig. 2).

A total of twelve studies reported the effect of the insular cortex on PSD. There studies conducted on an acute phase of stroke, investigated the potential lesion pattern related to PSD, and found that the right insular cortex is related to swallowing dysfunction and predictive for the development of dysphagia11,16,29. According to Hess et al., the MNI coordinates were X=−39, Y=−11, Z=10, and the voxels was 799.11 While another two studies, conducted on right hemispheric strokes, reported that associations were found in the left insular cortex.29,39 Furthermore, two studies conducted on supratentorial strokes by VLSM analysis, demonstrated that the anterior insular cortex was associated with the prognosis of PSD.17,35 The anterior insular cortex (MNI coordinates were X=39, Y=10, Z=20) was related to impaired oral intake 4 weeks after stroke, as reported by Galovic et al.,17 and affected 54% of voxels. Besides, Galovic et al. reported the anterior insular cortex was also related to the time before oral feeding,35 and found a significant difference in the anterior insular cortex (MNI coordinates are X=37, Y=10, Z=6) by the comparison between tube-dependency and no tube feeding patients, which affected 70% of voxels. The remaining six studies did not report the specific regions of the insular cortex.

  • It would be desirable to specify the presence of dysfagia before stroke-event in each study, if possible.  

Authors’ response: Thanks for your suggestion. We have revised Table1, Presence of dysphagia before stroke-event part has been added. To reduce the bias, the studies included need to exclude patients with dysphagia before stroke. Seventeen studies claimed that patients with dysphagia before stroke has been excluded, while the other seven studies remaining unclear.

Discussion: 

  • Please, change the format from line 224 to 262.

Authors’ response: Sorry about that, we have revised the manuscript.

  • The discussion in lack of information and i suggest to rewrite it at the light of the heterogeneity of  results. For example, the authors underline the presence of different lobar area involved in PSD, but focus their attention only on the insular cortex. Please, stress this point.

Authors’ response: Thanks for your suggestion. We have revised the Title, Objective, Result part of the manuscript. The aim of the present study was To investigate the relationship of lobar and deep brain regions with post-stroke dysphagia (PSD) (on Page 1). We found that The PSD-related lobar and deep brain regions included the insular cortex, frontal lobe, temporal gyrus, basal ganglia, postcentral, pretcentral, precuneus, corona radiate, periventricular white matter, etc. Among these brain regions, the insular cortex was most frequently reported (taking up 54.2%) in the included studies (on Page 2). The key point of the present study was “In lobar and deep brain regions, the insular cortex may be most relevant to PSD and aspiration, which may be a potentially promising target in the treatment of PSD (on Page 2). Therefore, the discussion part of the present study mainly focused on the insular cortex. Besides, the different lobar and deep brain regions related to PSD has also been described on Page 9-10.

  • Which are the clinical/rehabilitative implication of your work? please, add a paragraph to stress this point

Authors’ response: Thanks for your suggestion. We have revised the manuscript as follows (on Page 12): Several suggestions can be recommended according to our results. Firstly, the insular cortex participates in the various aspects of swallowing, and can be a potentially promising target for the treatment of aspiration. For example, noninvasive brain stimulation (NIBS) including repetitive transcranial magnetic stimulation (rTMS) and transcranial direct current stimulation (tDCS) is a practical technique, which has been proved effective and widely used to promote the recovery of PSD.56 However, the rTMS relies on accurate stimulation at specific brain regions to achieve clinical efficacy.57 Therefore, the insular cortex may become the target in the treatment of PSD and aspiration. Secondly, brainstem stroke is reportedly the main incentive of PSD,58 however, the lobar and deep brain regions also participate in the swallowing function regulation.59 After lobar and deep brain regions stroke, a comprehensive assessment may be needed to avoid serious complications according to our results.

Reviewer 3 Report

The paper is very well done, well written and very useful especially for the dysphagia sector. It would also be advisable to advertise it in the otolaryngology field to make it clear how important the brain  site of lesion, can be to identify dysphagia damage

Author Response

We are grateful to you for sparing your time in reviewing our paper and providing valuable comments. We appreciate the precious chance of major revision to improve our manuscript. We have revised the manuscript according to the suggestion.

Round 2

Reviewer 1 Report

Thank you for your answers

Reviewer 2 Report

Dear Authors,

Many compliments for your outstanding revision process. At the light of your revision, the paper is suitable for fully publication in Journal.

Best